# Drug Repurposing to Circumvent Immune Checkpoint Inhibitor Resistance in Cancer Immunotherapy

**DOI:** 10.3390/pharmaceutics15082166

**Published:** 2023-08-21

**Authors:** Kenneth K. W. To, William C. Cho

**Affiliations:** 1School of Pharmacy, Faculty of Medicine, The Chinese University of Hong Kong, Hong Kong SAR, China; 2Department of Clinical Oncology, Queen Elizabeth Hospital, Hong Kong SAR, China

**Keywords:** PD-1, PD-L1, drug repurposing, tumor microenvironment, drug resistance, immune checkpoint inhibitors, machine learning

## Abstract

Immune checkpoint inhibitors (ICI) have achieved unprecedented clinical success in cancer treatment. However, drug resistance to ICI therapy is a major hurdle that prevents cancer patients from responding to the treatment or having durable disease control. Drug repurposing refers to the application of clinically approved drugs, with characterized pharmacological properties and known adverse effect profiles, to new indications. It has also emerged as a promising strategy to overcome drug resistance. In this review, we summarized the latest research about drug repurposing to overcome ICI resistance. Repurposed drugs work by either exerting immunostimulatory activities or abolishing the immunosuppressive tumor microenvironment (TME). Compared to the de novo drug design strategy, they provide novel and affordable treatment options to enhance cancer immunotherapy that can be readily evaluated in the clinic. Biomarkers are exploited to identify the right patient population to benefit from the repurposed drugs and drug combinations. Phenotypic screening of chemical libraries has been conducted to search for T-cell-modifying drugs. Genomics and integrated bioinformatics analysis, artificial intelligence, machine and deep learning approaches are employed to identify novel modulators of the immunosuppressive TME.

## 1. Introduction

The recent advances in the development of immune checkpoint inhibitors (ICIs) have revolutionized cancer treatment. ICIs have been shown to induce a durable anti-tumor response and improve overall survival for patients bearing various cancer types. One of the hallmarks of cancer is its ability to evade immune surveillance and destruction. Under normal homeostasis, the immune checkpoints (including PD-1, CTLA-4, LAG-3, TIM-3, and TIGIT) function to curtail the over-stimulation of the immune system after exposure to an antigen and to avoid exacerbated immune responses. The delicate balance is maintained by the interaction of these inhibitory receptors expressed by immune cells with complementary co-stimulatory ligands expressed by antigen-presenting cells or myeloid cells [1]. In cancer patients, the inhibitory immune checkpoints on T cells could recognize and bind to their partner proteins in tumor cells, thereby suppressing the T cells and preventing the immune system from destroying the tumor.

The programmed cell death-1 receptor (PD-1) and its ligand programmed cell death ligand-1 (PD-L1) represent the two most extensively studied checkpoint proteins keeping immune responses in check. PD-1 is an inhibitory receptor expressed on activated T cells, B cells, and natural killer cells, which is engaged by its major ligand PD-L1 expressed on tumor cells to suppress the T-cell-mediated cancer killing effect [2]. Anti-PD-1/PD-L1 monoclonal antibodies were developed as ICIs to target this tumor immune evasion mechanism. They work by binding to the inhibitory PD-1 receptor on T cells and PD-L1 on tumor cells, respectively, thus disrupting the PD-1:PD-L1 interaction and reactivating the anti-tumor T-cell response. From clinical experience, cancer patients bearing high tumor mutational load, abundant pre-treatment tumor-infiltrating T cells, and high tumoral expression of pre-treatment PD-L1 are particularly responsive to anti-PD-1/PD-L1 therapy [3]. Similarly, other checkpoint molecules bind to their respective targets expressed on tumor cells to suppress the anti-tumor immune response. Thus, the fundamental principle of the various ICIs is to override immunosuppression and reactivate the adaptive immune response [4].

To date, the US Food and Drug Administration (FDA) has approved a few ICIs monoclonal antibodies (mAbs) for cancer therapy. They include (i) the anti-cytotoxic T-lymphocyte-associated antigen-4 (CTLA-4) mAb: Ipilimumab [5]; (ii) the anti-programmed death-1 (PD-1) mAbs: Nivolumab [6], Pembrolizumab [7], and Cemiplimab; (iii) the anti-programmed death ligand-1 (PD-L1) mAbs [8]: Atezolizmab, Avelumab, and Durvalumab; (iv) the anti-lymphocyte activation gene-3 mAb: Relatlimab [9]; and (v) the anti-T cell immunoglobulin and immunoreceptor tyrosine-based inhibitory motif domain (TIGIT) mAb: Tiragolumab [10].

The clinical success of ICIs in an ever-growing number of tumor types has fostered an immense interest in adopting the immunological approaches to treat cancer. Nevertheless, not all tumors respond to ICI therapy (primary resistance). Moreover, acquired resistance is severely hindering the clinical efficacy of ICIs. A few excellent and comprehensive reviews about the resistance mechanisms to cancer immunotherapy have been published recently [11,12,13]. The resistance mechanisms can be categorized into tumor-intrinsic and -extrinsic ones. Loss of antigen protein expression, absence of antigen presentation, and T-cell exhaustion are the most commonly reported tumor-intrinsic factors. On the other hand, the major tumor-extrinsic factors mediating ICI resistance include the presence of immunosuppressive cells (such as regulatory T cells (Tregs), myeloid-derived suppressor cells (MDSCs), and tumor-associated macrophages (TAMs)) within the tumor microenvironment (TME) and the absence of T cells with tumor antigen-specific T-cell receptors [14]. In order to maximize the clinical efficacy of ICIs, extensive research has been conducted with an aim to enhance tumor immunity.

## 2. Application of Drug Repurposing to Overcome ICI Resistance

Drug repurposing refers to the application of clinically approved drugs with well-characterized pharmacokinetic properties and known adverse effect profiles for a new indication. It used to be a serendipitous process when an off-target effect of a repurposed drug candidate was identified for a new medical use. More recently, numerous computational predictive tools and high-throughput screening methods have been used to expedite drug repurposing research. A few structure-based online tools, including DrugPredict [15], Protein–Ligand Interaction Profiler (PLIP) [16], and Protein Binding Sites (ProBis) [17], have been used to evaluate whether drugs with similar chemical structures are likely to interact with the desired protein targets for treating diseases beyond their original indications. On the other hand, some omics-based research tools, such as the Connectivity Map [18], the Library of Integrated Network-based Cellular Signatures [19,20], and the Drug Repurposing Hub [21], have been used to identify drugs with a similar transcriptional signature for drug repurposing. Recently, an integrated visual analytics tool called ClinOmicsTrail^bc^ has been developed to analyze clinical biomarkers and genomics/epigenomics/transcriptomics datasets to facilitate a holistic assessment of the repurposing use of targeted drugs as immunotherapeutic agents to treat breast cancer [22].

Non-oncology drugs have been combined with ICIs to boost anti-tumor immunity [23,24]. Repurposed drugs that do not exhibit direct cytotoxicity are expected to impose fewer toxicity problems on normal tissues. On the other hand, anti-cancer drugs not originally approved for cancer immunotherapy have also been used to enhance anti-tumor immunity, which is referred to as “soft repurposing” [25]. The repurposed drug candidates may act by either inducing an immunostimulatory effect or abolishing the immunosuppressive TME (Figure 1). As the repurposed drugs are already clinically approved, they represent more affordable treatment options for potentiating cancer immunotherapy and the novel combinations could be readily evaluated in the clinical setting. In the next section, various methods commonly used to identify drug repurposing candidates for circumvention of ICI resistance are discussed.

## 3. Methods for Identifying Drug Repurposing Candidates to Overcome ICI Resistance

### 3.1. Phenotypic Screening of Chemical Libraries for T Cell Modifying Drugs

A high-throughput screening method, which utilized a mouse model infected with the clone 13 variant of lymphocytic choriomeningitis virus (LCMV-CL13), was recently used to identify drug candidates capable of reversing T-cell exhaustion [27]. Infection with LCMV-CL13 was previously reported to maintain sustained expression of the inhibitory immune checkpoint receptors (including PD-1), the immunosuppressive cytokine interleukin-10, and suboptimal CD4 and CD8 T-cell activity [28]. In C57BL/6 mice infected with LCMV-CL13, virus-specific CD8^+^ T cells gradually lost their capacity to express IFN-γ, thus mimicking the immunosuppressive TME in vivo during T-cell exhaustion. In this high-throughput screening, a total of 19 positive hits was identified from the ReFRAME drug repurposing compound library (composed of FDA approved drugs (~35%) and investigational new drugs (INDs) currently or previously in clinical development (~65%)) that restored cytokine production and enhanced the proliferation of exhausted T cells [27].

### 3.2. Integrative Analysis of CRISPR/Cas9-Based Functional Screens for Repurposing

Clustered regularly interspaced short palindromic repeats (CRISPR)/Cas9 technology has been employed to identify novel therapeutic targets regulating tumor immunity in a high-throughput manner [29,30,31,32]. Both loss-of-function and gain-of-function screens could be conducted using CRISPR/Cas9 systems [33]. Cancer and immune cells’ co-culture systems mimicking the interaction between cancer and immune cells were used in in vitro screens [34]. On the other hand, in vivo screens conducted in animal models with an intact TME and a preserved immune system were expected to identify more clinically relevant molecular targets [34]. Most recently, comprehensive data collection of tumor immunity-associated functional screens has been conducted from publicly available data sets [35]. By integrating results from these CRISPR/Cas9 screens and multi-omics data from the TCGA pan-cancer cohort and ICI-treated datasets, novel molecular targets and transcriptome signatures useful for predicting response to cancer immunotherapy have been investigated.

Numerous experimental and clinical studies reported that the drug combination approaches could substantially increase the percentage of cancer patients responding to ICIs and subsequently lead to significant survival benefits. Li et al. adopted a signature matching approach to predict drugs that could be combined with ICIs to produce a synergistic anti-tumor effect [35]. By using data retrieved from screens involving ICI treatment, a list of ICI-enhancer genes and ICI-suppressor genes was generated, which was used as the query signature. On the other hand, drug signatures (i.e., drug-induced profiles of gene expression changes) were downloaded from the Connectivity Map datasets for matching with the query signatures (Figure 2). To foster clinical translation of the prediction, only clinically approved drugs or drug candidates that had passed phase I and II clinical trials were used in the analysis. Three signature matching methods, including XSum, KS, and RGES, were used to predict drug candidates that may be used to reverse the ICI resistance gene signature and, thus, are capable of enhancing ICI efficacy in a drug combination. Interestingly, among the top ten drug candidates identified, four of them (including irinotecan, quercetin, trifluridine, and resveratrol) were previously reported by preclinical and/or clinical studies to have immunomodulatory functions [36,37,38,39], thus supporting the reliability of the prediction.

### 3.3. Virtual Screening and Machine Learning to Identify Novel Modulators of the Immunosuppressive TME

As described above, IDO1 is the key enzyme catalyzing the degradation of tryptophan in the kynurenine pathway [40]. Importantly, the depletion of tryptophan in the TME leads to activation of the GCN2 kinase pathway, subsequently driving the differentiation of CD4^+^ T cells to Treg cells. Moreover, tryptophan metabolites (particularly kynurenine) produced from IDO1 catalysis could bind to the aryl hydrocarbon receptor (AhR) to induce Treg cells and a tolerogenic phenotype of dendritic cells [41]. It follows that IDO1 inhibition could revoke immunosuppressive TME. The adenosine A2A receptor represents another molecular target for immunomodulation [42]. An elevated level of adenosine in the TME is known to produce an immunosuppressive effect on NK and T cells. Thus, antagonism of the A2A receptor represents a novel strategy to potentiate ICI therapy [43]. Zhang et al. utilized a machine learning-based virtual screening method to identify IDO1 inhibitors from their in-house compound library [44]. They built naïve Bayesian (NB) and recursive partitioning (RP) models from a library of known IDO inhibitors and used 13 molecular fingerprints as descriptors to predict novel IDO inhibitors. From this work, three new IDO1 inhibitors were identified and their IDO inhibitory activity (at a low micromolar concentration range) was validated in cell-based assays. Interestingly, these IDO inhibitors belong to the tanshinone compound family, with a molecule scaffold derived from the traditional Chinese herb Danshen that is widely used for cardiovascular and cerebrovascular diseases.

Within the immunosuppressive TME, IL-6 plays a key role in inhibiting T-cell-mediated anti-tumor immunity [45]. Thus, the combination of IL-6 blockade and other anti-cancer therapies may result in an enhanced treatment response [46]. IL-6 is known to bind with GP130, which is a receptor subunit critical for the intracellular signaling regulation of cytokines. Using a combinatorial approach to target GP130, Chen et al. found that the osteoporosis drug bazedoxifene could block the interaction between IL-6 and GP130 and inhibit triple-negative breast cancer proliferation [47].

The status of macrophage polarization is known to regulate the TME. A high M1/M2 macrophage ratio is associated with reduced cancer susceptibility. A few targetable proteins involved in the regulation of macrophage polarization, including the cannabinoid receptor 2 [48] and the TRPV4 ion channel [49], have been reported. To this end, praziquantel (a clinically approved anthelmintic drug) [50] and capsaicin (a naturally occurring compound from chili peppers) [51] have been shown to alter macrophage polarization using machine and deep learning methods.

The pro-angiogenic vascular endothelial growth factor receptor VEGFR2 is also associated with the formation of an immunosuppressive TME. Inhibition of VEGFR2 was reported to prevent the recruitment of immature dendritic cells, MDSC, and Tregs, thereby abolishing the immunosuppressive TME [52,53]. Using a naïve Bayesian machine learning model, Kang et al. screened 1841 FDA-approved drugs for potential VEGFR2 inhibitors after employing a training set of 2598 inhibitors and 7764 decoys [54]. Three clinically approved drugs, flubendazole (an anthelmintic), rilpivirine (a HIV/AIDs drug), and papaverine (an alkaloid for treating smooth muscle spasm), were found to inhibit VEGFR2 with sub- to low-micromolar inhibitory activity.

In the past few years, numerous clinically approved drugs have been identified using the aforementioned methods with an aim to enhance ICI efficacy. In the next section, an updated account of representative drug candidates or drug classes for repurposing is described according to their mechanisms of overcoming ICI resistance.

## 4. Representative Repurposed Drug Candidates to Overcome ICI Resistance

### 4.1. Repurposed Drug Candidates Inducing Immunostimulatory Activities

Cytotoxic T lymphocytes (CTLs), commonly called CD8^+^ T cells, play a critical role in immune surveillance against pathogen (viruses and bacteria)-infected cells and malignant tumor cells in the body [55]. Pathogens and tumors are capable of upregulating the inhibitory checkpoint receptors on CTL surfaces to escape from the host’s immune surveillance, a process which is usually referred to as T-cell exhaustion [56]. ICI therapies (e.g., anti-PD-1 or anti-CTLA-4 mAbs) were designed to neutralize the inhibitory receptors (PD-1 or CTLA-4) on exhausted T cells, subsequently restoring the effector immune responses. To this end, the insufficient restoration of T-cell function could lead to ICI resistance [14]. A few classic chemotherapeutic drugs, targeted therapies, and epigenetic modifying drugs have been reported to induce anti-tumor immunity [57].

#### 4.1.1. Metronomic Chemotherapy (Also Called Low-Dose Chemotherapy)

Metronomic therapy refers to an alternate approach of chemotherapy administration where anti-cancer drugs are given at a reduced dose at regular and frequent time intervals. It was reported to produce a significantly greater anti-tumor immune response, less toxicity, and a reduced chance of therapeutic resistance [58]. Conventional chemotherapeutic drugs, including cyclophosphamide, etoposide, methotrexate, paclitaxel, and vinblastine, have been used in metronomic therapy for the treatment of various cancers. In fact, high-dose chemotherapy is known to target cancer cells but it is generally immunosuppressive. On the contrary, high-frequency metronomic chemotherapy is considered immunostimulatory due to its effect on the tumor stroma [58,59]. Metronomic therapy has been reported to reduce the abundance of immunosuppressive Tregs [60,61], promote maturation of antigen-presenting cells [62], and activate cytotoxic CD8^+^ T cells and NK cells [63]. Moreover, it was also shown to inhibit the immunosuppressive MDSCs within the TME [64,65]. In a murine model of lung adenocarcinoma, a combination of low-dose cyclophosphamide and oxaliplatin was reported to sensitize the tumor to ICI treatment by increasing the CTLs/Treg ratio [66]. Similarly, in another murine colorectal cancer model, low-dose oxaliplatin was found to potentiate the efficacy of anti-PD-L1 therapy by augmenting the activity of CTLs and dendritic cells [67]. In the CONFRONT phase I–II clinical trial, the combination of low dose cyclophosphamide and avelumab (anti-PD-1 mAb) was also shown to suppress the immunosuppressive effect of CD4^+^CD25^+^Foxp3^+^ Treg cells, which was accompanied by improved tumor-free survival in patients with metastatic head and neck cancer [68]. A few clinical trials investigating the combination of metronomic therapy of cytotoxic chemotherapeutic drugs with ICIs are summarized in Table 1.

#### 4.1.2. Molecular Targeted Drugs

##### Targeted Drugs with Anti-Angiogenic Activity

Cancer angiogenesis is a critical process that facilitates the formation of new and abnormal blood vessels to support tumor growth and metastasis. An effective anti-tumor immune response requires a series of events including the activation of T cells, recruitment of immune cells, and recognition and subsequent killing of cancer. To this end, inducers of angiogenesis are known to interfere with the activation, infiltration, and function of T cells. Moreover, the tumor vasculature is known to promote an immunosuppressive TME, which can be reversed using anti-angiogenic therapies [73]. Anti-angiogenic drugs work by inhibiting a few receptor tyrosine kinases, including vascular endothelial growth factor receptor (VEGFR), platelet-derived growth factor receptor (PDGFR), and fibroblast growth factor receptor (FGFR), which are involved in the angiogenic and proliferative pathways. Anti-angiogenic drugs are expected to activate anti-tumor immunity, whereas immunotherapy could exhibit an anti-angiogenic effect, thereby allowing the two drug classes to work synergistically in the treatment of cancer.

In a recent Phase 1b clinical trial (NCT03628521), the combination of anlotinib (a multikinase inhibitor against VEGFR, c-Kit, PDGFR, and FGFR) and sintilimab (anti-PD-1 mAb) gave rise to promising anti-tumor efficacy (median PFS = 15 months; ORR = 72.7%) without significant adverse effects in NSCLC patients [74]. Most recently, the combination of lenvatinib (a multikinase inhibitor against VEGFR1/2/3, FGFR, PDGFR, c-Kit, and RET) plus pembrolizumab (anti-PD-1 mAb) led to remarkably longer PFS (23.3 months versus 9.2 months, HR 0.42, respectively) than sunitinib monotherapy (standard first-line treatment) for advanced renal clear cell carcinoma (NCT02811861; Phase 3 CLEAR study) [75]. In another recent Phase 3 trial (KEYNOTE-426; NCT02853331), the combination of axitinib (a multikinase inhibitor against VEGFR, c-Kit, and PDGFR) and pembrolizumab showed longer PFS (15.4 versus 11.1 months, respectively) than single-agent sunitinib (first-line treatment) for advanced renal clear cell carcinoma [70]. In a retrospective study, the combination of regorafenib (a dual tyrosine kinase inhibitor against VEGFR2 and TIE2) and sintilimab (anti-PD-1 mAb) was also shown to produce better OS (13.4 versus 9.9 months), longer PFS (5.6 versus 4.0 months) and greater ORR (36.2% versus 16.4%) than regorafenib monotherapy (second-line treatment for advanced HCC) [76]. Table 1 shows a summary of recent clinical trials investigating the combination of ICIs and representative targeted drugs.

##### Other Small Molecule Tyrosine Kinase Inhibitors (TKIs)

Numerous small molecule TKIs have been developed to block specific intracellular oncogenic signaling pathways in cancer cells to suppress tumor proliferation and differentiation. They have become an indispensable part of modern precision oncology. Interestingly, a number of the targeted signaling pathways are also involved in the differentiation and activation of the immune cells. Therefore, TKIs also possess important immunomodulatory properties and could enhance the efficacy of ICI blockade therapy. Table 1 summarizes the combination effect of representative clinically approved TKIs and ICI therapy.

The mitogen-activated protein kinase (MAPK) pathway is an important oncogenic signaling cascade, which is composed of a series of signaling molecules including RAS, RAF, MEK, and MAPK. Besides regulating cancer survival and development, the MAPK pathway is also known to control anti-tumor immunity. In melanomas, the clinically approved BRAF inhibitor vemurafenib has been reported to induce T-cell antigen expression and stimulate a T-cell immune response [77]. The combination of a dual BRAF and MEK inhibitor with anti-PD-1 mAb could lead to more tumor infiltration of immune cells and better anti-tumor efficacy in a CD8^+^ T cell-dependent way [78]. Cobimetinib and trametinib are two recently approved MEK inhibitors for advanced melanoma and pediatric patients with low-grade glioma bearing a BRAF V600E mutation, respectively. Recently, it has been demonstrated that cobimetinib and trametinib could potentiate cancer immunotherapy by upregulating tumor antigen expression and presentation [79], promoting the production of IL-8 and VEGF, and enhancing the recruitment of immune cells to the tumor site [80]. Moreover, trametinib was also reported to upregulate MHC-class I expression by activating STAT3 signaling and promoting T-cell infiltration into tumor sites [81]. In a head and neck squamous cell carcinoma model, the combination of trametinib and anti-PD-1 mAb was shown to suppress tumor progression by enhancing CD8^+^ T cell activity and inducing long-term memory immune cells [82]. Table 1 summarizes the promising anti-tumor responses from combinations of these TKIs and ICI therapy in recent representative clinical trials.

Inhibition of the PI3K-Akt-mTOR signaling pathway is known to suppress glycolysis and potentiate the anti-cancer effect of chemotherapy [83,84]. Rapamycin (a prototype mTOR inhibitor) was clinically approved for treating various cancer types including metastatic renal cell carcinoma, pancreatic neuroendocrine cancer, and advanced breast cancer. Interestingly, the immunostimulatory effect of rapamycin has been shown in experimental models of infection and cancer, where rapamycin promoted the production of memory CD8^+^ T cells [85]. Moreover, rapamycin was also reported to exhibit cytotoxic effects on γδ T cells [86]. In mice bearing MOC1 (oral cavity) tumors, the combination of rapamycin and anti-PD-L1 mAb was shown to prolong survival more than the individual treatment alone [87]. The drug combination was found to increase tumor infiltration and activation of antigen-specific CD8^+^ T cells. Further ex vivo analysis of the CD8^+^ tumor-infiltrating lymphocytes (TILs) revealed that rapamycin enhanced the production of IFN-γ by the CD8^+^ TILs [87]. Several clinical trials are ongoing to investigate the beneficial anti-tumor effect of the combination of rapamycin and ICI blockade therapy (NCT02890069, NCT04348292, and NCT03190174).

The ErbB/HER family of protein tyrosine kinases is among the most extensively studied cell signaling families in biology. There are four members including ErbB1/epidermal growth factor receptor (EGFR), ErbB2/HER2, ErbB3, and ErbB4. Constitutive activation of these protein tyrosine kinases drives tumorigenesis and the development of various cancer types. Numerous small molecule TKIs and mAbs targeting EGFR and HER2 were developed and approved for cancer treatment. Besides affecting cancer cell signaling, EGFR TKIs were also known to affect T-cell tumor antigen recognition, T-cell activation, and tumor infiltration of immune cells [88]. A number of clinical trials were conducted to investigate the combination of small molecule EGFR TKIs (including gefitinib, erlotinib and osimertinib) in NSCLC patients [89,90,91,92] (Table 1). In these studies, the drug combinations were found to significantly increase the adverse events (mainly hepatotoxicity). In contrast, the combination of EGFR mAbs and ICIs in various cancer types (including CRC, HNSCC, and NSCLC) demonstrated promising clinical efficacy (improved median PFS) without significant adverse effects (Table 1) [93,94,95,96]. Detailed mechanistic studies showed that the EGFR mAb cetuximab facilitated dendritic cell priming to augment anti-tumor immunity [97]. Moreover, cetuximab was also reported to promote natural killer cells-mediated antibody-dependent cellular cytotoxicity and complement-dependent cytotoxicity to increase its anti-tumor effect in combination with ICI therapy [98,99].

##### Cyclin-Dependent Kinase (CDK) Inhibitors

Cell cycle dysregulation is a well-recognized hallmark of cancer. CDKs are protein kinases regulating cell cycle progression, gene transcription, and various other cellular functions [100]. The overexpression or dysregulation of CDKs plays an important role in driving the unlimited proliferation of cancer. In particular, the overexpression of cyclin D1 (the binding partner of CDK4/6) and the loss of function of p16^INK4a^ (endogenous CDK4/6 inhibitor) lead to abnormal function of CDK4/6, thus compromising the G1/S cell cycle checkpoint in cancer [101]. A few clinically approved CDK4/6 inhibitors, including palbociclib, ribociclib, and abemaciclib, demonstrate promising anti-tumor activity in hormone receptor-positive and HER2-negative breast cancer [102]. Interestingly, recent studies in melanoma suggested that CDK4/6 inhibitors also exhibited complementary immunotherapeutic activity for cancer treatment. Palbociclib was reported to enhance the anti-tumor activity of anti-PD-1/PD-L1 ICIs by stimulating the tumor production of type III interferons (IFNs) and increasing tumor cell surface expression of MHC class I proteins [103]. In the murine breast cancer model, palbociclib was also shown to reduce the PD-L1 expression and increase the tumor cell production of CXCL10 and CXCL13 chemokines, thereby increasing lymphocyte recruitment to the TME [104,105]. Moreover, CDK4/6 inhibitors were also found to diminish Treg proliferation and enhance the effector T cell activity by downregulating the nuclear factor of activated T cells (NFAT, which regulates Treg transcription) [106]. Furthermore, CDK4/6 inhibitors were also reported to promote the formation of stem or memory-like cytotoxic CD8^+^ T cells, thus leading to a sustained anti-tumor response to ICIs [107,108]. In a recent phase 1/2 clinical trial (NCT02778685), the combination of palbociclib, pembrolizumab, and letrozole was well-tolerated and produced a promising objective response rate (ORR) of 56% as the first-line treatment for HR-positive metastatic breast cancer [109]. The combination of abemaciclib and pembrolizumab was evaluated in another phase 1b trial for patients with NSCLC and HR+/HER2− breast cancer (NCT02779751). The abemaciclib–pembrolizumab combination showed a good ORR of 14.3% with a tolerated safety profile in HR+/HER2− breast cancer patients [110]. However, the drug combination gave rise to more severe toxicity than the two individual drugs alone in NSCLC patients [111]. More clinical investigation will be needed to elucidate whether there is cancer-type selectivity for the beneficial anti-tumor efficacy from the combination of CDK4/6 inhibitor and ICI treatment.

##### DNA Damage Response Inhibitors (DDRIs)

DDRIs have been developed to target cancers with existing defects in DNA repair [112]. Poly(ADP-ribose) polymerase (PARP) is critical in DNA repair pathways. Tumors with defective homologous recombination, particularly *BRCA* mutation, are susceptible to PARP inhibitors [113]. Apart from being used as a monotherapy in cancer treatment, DDRIs were also reported to produce synergistic effects with other anti-tumor treatment modalities or reverse acquired treatment resistance [112].

Classical chemotherapeutic drugs and DDRI are known to increase the load of DNA damage in cancer cells and trigger an innate immune response [114]; thus, theoretically they could be combined with cancer immunotherapy to give rise to a better anti-tumor effect. However, classical chemotherapeutic drugs are not selective and they kill both cancer cells and immune cells. Therefore, they are generally poor candidates for combination with immunotherapies. On the other hand, DDRIs were designed to target the tumor-specific defects and they are less cytotoxic to healthy tissues. PARP inhibitors were first identified as synthetic lethal interactors with *BRCA2* mutations [115]. They also demonstrated a similar synthetic lethal interaction with any mutation that results in defective homologous recombination repair (HRR). Olaparib, the first-in-class PARP inhibitor, was shown to increase DNA damage which subsequently induced an innate immune response through the cGAS–STING pathway (cyclic-GMP-AMP synthase cGAS-Stimulator of Interferon genes) [116]. PARP inhibitors were shown to improve immune recognition, which could be further enhanced with ICIs [117,118]. Interestingly, this effect was independent of the functional status of HRR. Numerous clinical studies have been conducted to evaluate the combination of olaparib and ICIs [119,120] (Table 1). Initial findings from a phase 1/2 clinical trial in recurrent platinum-resistant ovarian cancer showed that the niraparib–pembrolizumab combination exhibited promising anti-tumor activity with an ORR of 18% (NCT0265788) [121]. Another recent phase 2 trial also reported modest anti-tumor efficacy from a combination of olaparib and durvalumab in advanced prostate and ovarian cancer without significant adverse effects [122,123].

#### 4.1.3. Epigenetic Drugs

At an advanced stage of tumor development, T cells within the TME are known to acquire a terminal exhaustion state following various mechanisms, including DNA methylation, and the process is generally irreversible [124]. To this end, the inhibition of de novo DNA methylation has been shown to potentiate anti-PD-1/anti-PD-L1 blockade therapy [125,126]. Azacitidine (a DNA methyltransferase (DNMT) inhibitor approved for acute myeloid leukemia (AML)) was shown to upregulate PD-1 and IFNγ signaling. Importantly, azacitidine was reported to produce promising anti-tumor efficacy (overall response rate = 33%) when used in combination with nivolumab (anti-PD-1 mAb) in refractory AML patients with manageable side effects [127]. Another DNMT inhibitor (decitabine, clinically approved for myelodysplastic syndromes and AML) was also reported to achieve a higher rate of complete remission (79% in drug combination versus 32% for camrelizumab alone) and produce a long-term survival benefit (median PFS: 35 months in drug combination vs. 15.5 months with camrelizumab alone) when used in combination with camrelizumab (anti-PD-1 mAb) in relapsed Hodgkin lymphoma [128]. The use of decitabine was also reported to enhance the anti-tumor efficacy of adoptively transferred CAR T cells with significantly increased production of cytokine [129]. DNA hypomethylating agents were also reported to elevate the expression of cancer-specific antigens and major histocompatibility complex (MHC), which subsequently promoted immunologic recognition of cancer cells and anti-tumor immunity [130].

Histone deacetylase inhibitors (HDACIs) are another class of potent epigenetic drugs with proven anti-tumor effects in both hematological and solid cancers. They work by inducing histone acetylation at lysine residues, thereby opening up chromatin configuration to regulate gene expression. Interestingly, HDACIs are also known to alter the expression of immune system upregulators, including MHC and costimulatory molecules, which subsequently affects antigen presentation and T cell activation [131,132]. Class I HDACIs are known to enhance the activity of natural killer and CD8^+^ T cells. Class II HDACIs could target Tregs, whereas HDAC6 inhibitors were reported to stimulate naïve T cells. In a syngeneic murine model, HDACI treatment was shown to upregulate PD-L1, retard tumor progression, and lead to longer survival [133]. In a recent phase 1/1b clinical trial investigating the combination of vorinostat and pembrolizumab in patients bearing metastatic NSCLC, partial response and stable disease were observed in 4 (13%) and 16 (53%) subjects without any dose-limiting toxicities [134].

#### 4.1.4. Drugs Promoting M1 Macrophage Polarization

Tumor-associated macrophages (TAM) within the TME play a critical role in determining tumor development, chemoresistance, immune evasion, and metastasis [135]. Depending on the activation status of TAMs, they can either promote or suppress tumor growth. The M1 (or classically activated) macrophages are endowed with anti-tumor activity, whereas the M2 (or alternatively activated) macrophages promote cancer proliferation [136]. The depletion of total TAMs, which aims to counter the tumor promotion effect of the M2 TAMs, has been investigated as an effective anti-tumor strategy [137]. More recently, the reprogramming of TAMs towards a tumoricidal M1 phenotype has been proposed as a novel approach to enhance cancer immunotherapy [138,139].

Mallardo et al. recently conducted a retrospective study on 121 patients with stage IIIb–IV metastatic melanoma [140]. They found that the concomitant use of anti-PD-1 immunotherapy and cetirizine (an antihistamine) was associated with improved PFS, OS, ORR, and DCR in patients naïve to ICI therapy. Data from the transcriptomic analysis revealed that patients receiving cetirizine and naïve to anti-PD-1 therapy had a higher expression of interferon-related genes (including *FCGR1A*, *CCL8*, *IFIT1*, *IFIT3*, and *RSAD2*) compared to the baseline. In fact, cetirizine possesses antihistamine and anti-inflammatory effects and it is known to enhance IFN-γ production by peripheral blood monocytes [141]. In another mouse study, antihistamine treatment was shown to retard the growth of colorectal cancer and enhance the cytokine-induced immune response [142]. Taken together, cetirizine may potentiate the anti-tumor efficacy of anti-PD-1 ICI by promoting the M1 polarization of TAMs via the IFN-γ pathway.

### 4.2. Repurposed Drug Candidates Abolishing Immunosuppressive TME

The TME recruits numerous immunosuppressive cells (including Treg, TAM, MDSC, and CAF), which collectively lead to dysregulation of the immune checkpoints, inhibit tumor antigen presentation, and suppress T-cell activation (Figure 1). The immunosuppressive TME inhibits the killing of tumor cells by CD8^+^ T cells and NK cells, thereby promoting immune evasion. Drugs capable of inhibiting the intrinsic oncogenic signals may be used to modulate the immune response by overcoming immunosuppression.

#### 4.2.1. Angiotensin II Receptor Blockers (ARBs)

The renin–angiotensin–aldosterone system is a crucial hormone system in the body that regulates blood pressure and fluid balance. When the systolic blood pressure falls or there is a rapid drop in blood volume, renin will be secreted by the kidneys into the bloodstream to restore blood pressure and promote fluid retention. ARBs are drugs that target this renin–angiotensin–aldosterone system to treat hypertension and congestive heart failure. Interestingly, angiotensin II is also known to promote an immunosuppressive TME and induce cancer-associated inflammation [143]. Thus, ARBs have been investigated for the reprogramming of the TME and overcoming ICI resistance [144] (Figure 3).

Hypoxic tumor cells were shown to produce angiotensin II locally at a high concentration [145]. In a murine colorectal cancer model, the clinically approved ARB (candesartan) was reported to inhibit angiotensin II signaling, thereby impairing the suppressive TME and inducing infiltration of CD8^+^ T cells to enhance the anti-tumor response to ICI therapy [143]. Another commonly used ARB (valsartan) was also shown to reduce the production of immunosuppressive factors (such as IL-6 and VEGF) from MDSC and macrophages [146]. Valsartan was also reported to decrease CXCL12 and NOS-2 expression in cancer-associated fibroblasts, thereby augmenting the anti-tumor response to immunotherapy. In a murine colon cancer-bearing mouse model, the combination of valsartan and anti-PD-1 mAb was shown to abolish the immunosuppressive TME and potentiate the CD8^+^ T-cell-mediated anti-tumor response. Similarly, in mice inoculated with malignant melanoma, valsartan was found to reduce the production of the CC motif chemokine ligand CCL5 from fibroblasts, increase tumor-infiltrating T cells, and decrease regulatory T cells, which was accompanied by an increase in tumor antigen-specific T-cell responses [147]. Moreover, the valsartan-anti-PD-1 mAb combination was shown to elicit significantly more tumor growth inhibition than anti-PD-1 mAb monotherapy. It is noteworthy that ARBs can only be applied at limited doses for cancer therapy because they lead to hypotension due to angiotensin II inhibition [148]. Recently, TME-activated ARB nanoconjugates (TMA-ARBs) have been designed that could preferentially accumulate in tumors and exert their effects on anti-tumor immunity [149]. In these ARB nanoconjugates, the ARB molecule (valsartan) is chemically linked to polymers that are sensitive to the acidic tumoral pH. The TMA-ARBs remain intact in blood circulation, but they break down readily in tumors to release the active ARBs and achieve a high drug concentration. Thus, the specific tumor-site activation of TMA-ARBs enhanced the TME modulating effect of ARBs without a significant impact on blood pressure control. Importantly, in mouse models bearing primary as well as metastatic breast cancer, TMA-ARBs were shown to attenuate immunosuppression and potentiate anti-tumor T-cell activity, thereby enhancing anti-tumor response to anti-CTLA4 and anti-PD-1 combination ICI therapy.

#### 4.2.2. Non-Steroidal Anti-Inflammatory Drugs (NSAIDs)

Prostaglandin (PGE2) is a potent inflammatory mediator that is converted from arachidonic acid by the action of cyclooxygenase2 (COX2). It plays an important role in conferring resistance to cancer immunotherapy [150]. PGE2 is known to suppress the conversion of the T helper 1 (Th1) to the T helper 2 (Th2) phenotype, thereby inhibiting cytotoxic T-cell formation and promoting cancer proliferation [151]. Moreover, PGE2 also promotes the propagation of the immunosuppressive Treg2, MDSC, and M2 macrophage population [152]. Therefore, COX inhibitors/NSAIDs have been hypothesized to sensitize tumor cells to immunotherapy (Figure 4).

Aspirin is a classical anti-inflammatory drug and its anti-cancer activity has been reported in numerous cancer types [153]. The combination of aspirin and anti-PD-1 mAb has been shown to produce a higher anti-tumor immune response than anti-PD-1 monotherapy alone in mice bearing the CT26 colon cancer xenograft [154]. In a large retrospective chart review of 500 NSCLC patients on PD-1- or PD-L-directed immunotherapy, daily aspirin use with anti-PD-L1 was associated with a higher likelihood of achieving complete remission (adjusted odds ratio = 1.85) [155]. Interestingly, the adjuvant use of aspirin was found to have a stronger association with survival in NSCLC patients with low tumoral PD-L1 expression. This observation suggested that PD-L1 expression may be used as a potential biomarker to select patients for a combination of anti-PD-L1 ICI and aspirin for enhanced anti-tumor efficacy. The combination of aspirin and anti-PD-L1 or anti-CTLA4 therapies in various cancer types is also under investigation in a few ongoing clinical trials (NCT02659384 (phase 2 in ovarian cancer); NCT03245489 (phase 1 in head and neck cancer); NCT03396952 (phase 2 in melanoma)).

Celecoxib is a COX-2 inhibitor and it was shown to promote the conversion of tumor-associated macrophages from an immunosuppressive M2 to a tumoricidal M1 phenotype in Apc^Min/+^ mouse polyps [156]. In a PD-1/PD-L1-resistant B16F10-R tumor mouse model, the combination of celecoxib and pembrolizumab (anti-PD-1 mAb) was shown to effectively reverse the drug resistance by increasing the number of immune cells infiltrating the TME [157].

Naproxen is a nonselective inhibitor of both COX-1 and COX-2. In APC Min mice, both naproxen and celecoxib were found to inhibit polyp growth and reduce PD-L1 expression in intestinal tumor cells. The decrease in PD-L1 expression was associated with an influx of CD8^+^ T cells into the colon polyps (which appeared to be more significant for celecoxib than naproxen). More in-depth investigation revealed that NSAID regulation of PD-L1 was dependent on COX-2 expression. In the murine colon cancer cell line MC38, PD-L1 expression was reduced by 86% in COX-2-silenced cells, whereas there was a negligible effect after COX-1 silencing. Furthermore, naproxen and celecoxib were also shown to reduce PD-1 and LAG3 expression on regulatory T cells, presumably inhibiting the immunosuppressive TME to enhance the anti-tumor efficacy of anti-PD-1/PD-L1 therapy [158].

#### 4.2.3. Drugs Modulating Metabolic Pathways to Reprogram the Immunosuppressive TME

TME is highly hypoxic and nutrient-deprived due to aggressive cancer growth and inadequate vascular supply. Cancer cells and tumor-infiltrating immune cells are subjected to metabolic stress, thus impairing anti-tumor immune responses. A few excellent reviews have been published on this topic in recent years [159,160,161]. The repurposing of drugs capable of modulating cancer metabolism may enhance the anti-tumor efficacy of ICIs through metabolic reprogramming of the TME.

##### Targeting Glucose Metabolism to Enhance Anti-Tumor Efficacy of ICI Therapy

The Warburg effect of aerobic glycolysis is a characteristic metabolic hallmark of cancer, which is adopted instead of the more efficient oxidative phosphorylation as the major mode of glucose metabolism to support rapid cancer proliferation and malignant progression [162]. As a consequence, a high level of lactic acid is generated in the extracellular milieu, which mediates the immunosuppressive properties of TME by inhibiting the proliferation of cytotoxic T lymphocytes [163,164,165,166,167]. A few therapeutic approaches have been proposed to inhibit lactic acid production by targeting lactate dehydrogenase (LDH) and monocarboxylate transporter (MCT) to regulate the acidic TME [168]. LDH is the crucial enzyme catalyzing the conversion of pyruvate to lactate and the regeneration of NAD^+^ during glycolysis, which is usually highly expressed in cancer cells [169]. Galloflavin (synthesized from gallic acid) has been shown to inhibit LDH and reduce lactate levels [170]. However, it was also shown to suppress IFN-γ production by T cells [171]. A differential effect of LDH inhibitors on cancer and immune cells is therefore preferred when they are combined with cancer immunotherapy. Furthermore, inhibition of the lactate transporters MCT1-4 has also been proposed to inhibit the formation of acidic TME [172]. To this end, a few clinically approved drugs (including thalidomide, lenalidomide, and pomalidomide) have been identified as novel MCT inhibitors [173]. In particular, lenalidomide was also reported to promote the secretion of interleukin-2 (IL-2) and IFN-γ from T cells [174], which may allow its dual role in suppressing cancer proliferation and activating T-cell function. Diclofenac is a non-steroidal anti-inflammatory drug. In a murine glioma model, it was found to inhibit lactic acid formation and counteract the local immune suppression in TME, thereby inhibiting tumor growth [175,176].

Interestingly, the use of bicarbonates to neutralize the acidity within the TME has been proposed to potentiate the efficacy of cancer immunotherapy [177]. Oral administration of bicarbonate has been reported to inhibit melanoma growth when combined with anti-PD-1 ICI therapy and it also prolonged survival upon combination with adoptive T-cell transfer [178]. Moreover, esomeprazole (a proton pump inhibitor) was also shown to neutralize the acidic pH of TME, thereby enhancing the anti-tumor efficacy of cytotoxic T lymphocytes and NK cells [179]. However, the results from a recent meta-analysis revealed that proton pump inhibitors led to significantly worse PFS and OS in advanced-cancer patients treated with ICIs, probably by modulating the diversity of the gut microbiota [180]. Further investigation is needed to elucidate the clinical effect of proton pump inhibitor–ICI combinations.

Antihyperglycemic drugs have been repurposed to modulate the glucose metabolism of cancer cells and reprogram the immunosuppressive TME to enhance the anti-tumor efficacy of cancer immunotherapy. Metformin, a widely prescribed anti-hyperglycemic drug, is the most extensively studied in combination with ICI therapy. The fact that metformin consumption is associated with reduced tumor incidence in diabetic patients has attracted research attention to investigate its direct and indirect anti-tumor effects in various cancer types [181]. Various mechanisms were reported to contribute to the beneficial combination of metformin and ICI therapy in preclinical studies [182,183]. In particular, metformin has been combined with cancer immunotherapy to overcome resistance by regulating the hypoxic TME (Figure 5). Hypoxia in solid tumors is known to promote infiltration of various immunosuppressive cells (including MDSC, TAMs, and Tregs), inhibit proliferation and differentiation of cytotoxic T cells, and facilitate the stem cell-like phenotype that is resistant to CD8^+^ T cell-mediated cytotoxicity, which collectively contributes to resistance to cancer immunotherapy [184,185]. To this end, metformin has been reported to modulate the TME by diminishing the intra-tumoral accumulation of myeloid-derived suppressor cells (MDSCs) [186] and by abolishing the immunosuppressive phenotype [187,188]. Moreover, metformin can also attenuate tumor hypoxia by inhibiting the mitochondrial complex, thus reducing oxidative phosphorylation and leading to a better anti-tumor response to PD-1 blockade and δ T cell immunity [182,189,190]. Furthermore, metformin was also reported to increase CD8 T-cell infiltration and survival in hypoxic tumor regions, and produced a synergistic anti-tumor effect with cyclophosphamide to potentiate the efficacy of ICI or adoptive cell therapy in various tumor models [191].

Moreover, metformin was shown to decrease MDSC accumulation in tumors but increase proliferation and cytokine production from tumor-infiltrating CD8^+^ T cells. In a mouse melanoma model, metformin was shown to inhibit cancer-cell glycolysis but promote oxidative phosphorylation and cytokine secretion of CD8^+^ T cells [182]. On the other hand, metformin was also reported to induce phosphorylation of PD-L1 (at Ser195) by activating AMPK, subsequently leading to aberrant glycosylation and accelerated degradation of PD-L1 in the endoplasmic reticulum. This finding has advocated the combination of metformin and anti-CTLA-4 mAb as an alternative treatment strategy that resembles the dual immune checkpoint blockade regimen of anti-PD-1/anti-CTLA-4 combination [183].

While metformin alone did not appreciably reduce the tumor burden in aggressive tumors, the combination of metformin and anti-PD-1 blockade was reported to enhance the cytotoxic T-cell function and promote tumor clearance and regression [182]. Fueled by the encouraging anti-cancer activity of the metformin–ICI combination in preclinical studies, several clinical trials have been initiated to investigate their clinical anti-tumor efficacy in different cancer types (Table 1). However, in a recent meta-analysis (including eight studies), the concomitant use of metformin in cancer patients receiving ICIs was not significantly associated with clinical benefits [192]. It will be interesting to examine whether there could be cancer type-specific benefit achieved by a metformin–ICI combination.

##### Targeting Amino Acid Catabolism to Potentiate Cancer Immunotherapy

Many cancers exhibit metabolic addiction to specific amino acids [193]. They could become dependent on either an exogenous source or augmented de novo synthesis of the amino acids. Therefore, the combination of amino acid starvation with other anti-cancer therapies has been proposed to improve the treatment outcome [194]. In particular, tryptophan is crucial for the maintenance of an immunosuppressive phenotype in various cancer types [195]. Indoleamine 2,3-dioxygenase 1 (IDO1) is a rate limiting catabolic enzyme that catalyzes the degradation of tryptophan. The depletion of tryptophan by IDO1 leads to the suppression of cytotoxic T cells, differentiation of naïve T cells to immunosuppressive Treg cells, induction of immunosuppressive MDSC activity, and recruitment of tumor vasculature [196]. The genetic silencing of IDO1 has consistently been reported to reduce the abundance of Treg cells and potentiate cytotoxic T cells [197], and enhance the anti-tumor immunity in metastatic hepatocellular carcinoma [198]. IDO1 is considered a useful therapeutic target for drug discovery in cancer immunotherapy [199,200]. Indeed, the targeted chemotherapeutic drug for leukemia, imatinib, has been shown to enhance anti-tumor immunity by activating cytotoxic T cells and suppressing Tregs in an IDO1-dependent manner [201]. The combination of imatinib and anti-CTLA-4 mAb is currently under investigation in clinical trials for gastrointestinal stromal cancer [202].

Glutamine is an important source of carbon and nitrogen necessary for the anabolic metabolism of cancer cells [203]. As glucose is metabolized to lactic acid by glycolysis in cancer cells, glutamine is used to generate metabolic intermediates via the tricarboxylic acid (TCA) cycle to support cancer metabolism and proliferation. This specialized metabolism of cancer cells creates acidic, hypoxic, nutrient-depleted, and immunosuppressive TME that opposes anti-tumor immune responses. Therefore, the inhibition of glutamine metabolism is expected to inhibit cancer growth and also restore anti-tumor immunity. In fact, accumulating evidence suggests that inhibitors of glutamine metabolism, such as V-9302 and 6-diazo-5-oxo-L-norleucine (DON) derivatives, give rise to a pronounced anti-tumor effect when used in combination with ICIs by remodeling the immunosuppressive TME [204,205]. In murine models, V-9302 was shown to induce tumoral PD-L1 expression and augment immune evasion [206]. The blockade of glutamine metabolism reduced cellular GSH level and endoplasmic reticulum Ca^2+^-ATPase (SERCA) glutathionylation, subsequently leading to a reduced SERCA activity. The elevation of cytosolic Ca^2+^ level activated calcium/calmodulin-dependent protein kinase II, thereby leading to aberrant NF-kB signaling and higher PD-L1 expression. On the other hand, DON derivatives inhibit a broad range of glutamine-requiring enzymes including glutaminase [205]. Effector T cells responded to DON derivatives by upregulating oxidative metabolism and adopting a highly activated phenotype, thereby giving rise to increased anti-tumor immunity.

### 4.3. Repurposing Traditional Chinese Medicine (TCM) to Potentiate ICI Efficacy and Overcome Drug Resistance

Accumulating evidence indicates that TCM is a promising strategy for the treatment of cancer [207]. TCM has been combined with other anti-cancer treatment modalities, including radiotherapy, chemotherapy, and targeted therapy, with an aim to enhance the treatment efficacy and alleviate adverse drug effects. In recent years, numerous studies also suggest that TCM could modulate the TME to reverse ICI resistance, potentiate clinical efficacy, and reduce the severity of immune-related adverse events (irAEs) [208,209]. The repurposing use of a few representative TCM drugs in potentiating ICI efficacy and their mechanisms of resistance circumvention are described below.

Artemisinin is derived from extracts of sweet wormwood (*Artemisia annua*) and is well established for the treatment of malaria [210]. Recent research suggests that Artemisinin also possesses a direct anti-cancer effect by releasing excessive reactive oxygen species and indirectly by regulating the immune cell response against cancer [211,212]. In various tumor-bearing mouse models, artemisinin was reported to inhibit the accumulation and function of MDSCs in the TME by promoting an M2-to-M1 macrophage transition [213]. The switch to the anti-tumor M1-like phenotype was associated with the inhibition of the PI3K/AKT, mTOR and MAPK signaling pathways by artemisinin. Importantly, artemisinin was also shown to potentiate the anti-tumor activity of PD-L1 blockade therapy in tumor-bearing mice by promoting the tumor infiltration and proliferation of cytotoxic T cells. Cryptotanshinone is an active ingredient of the TCM herb Danshen, which is generally used for the treatment of cardiovascular and cerebrovascular conditions [214]. In a lung cancer model with high expression of the chemokine ligand 9 (CXCL9), CXCL11, and granzyme B, cryptotanshinone was shown to increase the infiltration of CD8^+^ T and CD4^+^ T cells in the TME to enhance the anti-tumor effect of PD-1 blockade therapy [215]. *Salvia miltiorrhiza* is a popular medicinal plant used for the treatment of coronary heart diseases and cerebrovascular diseases [216]. The active ingredient in *Salvia miltiorrhiza* (Salviaric acid B) was shown to potentiate CD8^+^ T cell infiltration in the TME to increase the efficacy of anti-PD-L1 mAb in a breast cancer animal model, which was also accompanied by an endothelial protective effect and normalization of vascular function [217]. *Astragalus membranaceus* has been traditionally used in TCM to reduce blood sugar, reduce blood lipids, regulate immune function, and elicit anti-cancer and anti-viral effects [218]. In a mouse melanoma B16 model, the intranasal administration of *Astragalus membranaceus* polysaccharides was shown to activate DCs in the mesenteric lymph nodes and stimulate NK and T cells to potentiate the anti-tumor effect of anti-PD-L1 mAb [219]. A Ginseng-derived nanoparticle preparation was reported to reprogram TAMs, thus leading to the recruitment of CD8^+^ T cells to the TME and producing a synergistic anti-tumor effect with anti-PD-1 mAb [220]. *Ailanthus altissima* is a medicinal plant with a long history of use in China for the treatment of ascariasis, diarrhea, spermatorrhea, and gastrointestinal disorders [221]. Ailanthone (an active component extracted from *Ailanthus altissima*) was reported to produce a synergistic anti-cancer effect in a melanoma model by inhibiting the infiltration of the immunosuppressive Tregs in the TME [222].

TCM is also known to enhance the anti-tumor effect of ICI via regulating the gut microbiota. The gut microbiota refers to the collection of numerous microbes inhabiting the human intestinal epithelium in the gastrointestinal tract. The microbiota maintains complex interactions with the intestinal cells and resident immune cells through their metabolites. Short-chain fatty acids, purine metabolites, and tryptophan derivatives are well-known gut microbiota-derived metabolites that regulate immune responses and modulate the anti-tumor efficacy of ICI [223,224]. Ginseng polysaccharides have been shown to sensitize lung cancer cells to PD-1 blockade therapy by increasing the abundance of the Gram-negative bacteria *Muribaculum* in the gut [225]. The popular TCM formulation Gegen Qinlian decoction was also reported to enhance the efficacy of ICI therapy in a colon tumor model by enriching the intestinal content of *Bacteroides acidifaciens* and *Peptococcaceae* [226]. The interaction between the gut microbiota and TCM is believed to contribute to the beneficial anti-tumor effect from the combination of ICIs and TCM [227]. The gut microbiota may transform the TCM components more readily to the active compounds. Alternatively, the TCM drugs may regulate the gut microbiota to increase the anti-tumor immune response and lessen the impact of irAEs [228].

## 5. Challenges and Perspectives

In recent years, ICI therapy has emerged as an effective therapeutic strategy to produce durable anti-tumor responses and survival benefits in a wide variety of cancer types. Despite the great innovation of ICI therapy, the response rate is low (inherent resistance). The responding patients will eventually relapse because of adaptive resistance. To unravel the full potential of ICI therapy, the mechanism driving the de novo and adaptive resistance is a research area of intensive investigation. The combination of ICIs and other treatment modalities has been investigated for circumventing the resistance problem with promising outcome. In particular, non-oncology drugs were repurposed to boost anti-tumor immunity, which work by either inducing an immunoreactive effect or abolishing the immunosuppressive TME.

A systematic workflow and streamlined strategy to identify potential repurposed drugs for potentiating ICI therapy is highly desirable. Some repurposed drug candidates may be able to enhance the anti-tumor efficacy of cancer immunotherapy, but only at a dose substantially higher than the one used in their original indications. In the case of de novo drug discovery, the structure of the lead compounds may be modified to enhance the potency and improve other pharmacokinetic properties. However, for research in drug repurposing, the positive candidates often cannot be replaced without compromising the desired outcome. Moreover, the higher dose needed for drug resistance circumvention may not be achievable in vivo due to either limited solubility or pharmacokinetic constraints. While the adverse effect profile of the repurposed drugs is available upon original approval, unexpected toxicity is still possible, especially at high doses. Moreover, the possibility of increased severe irAEs from ICI therapy drug combination also raised safety concerns. Novel tools are needed to reliably predict potential adverse effects in patients receiving a combination of ICIs and repurposed drugs [229].

Nanoparticle (NP)-based formulations have been used to deliver ICIs with an aim to increase their specificity, decrease toxicity, and potentiate the immunostimulatory effect [230]. NPs could be designed with a different size, geometry, and composition to facilitate a localized and controlled ICI release, and to protect ICI stability after administration into the body. Nanocarriers are preferentially delivered into tumors using passive targeting via enhanced permeation and retention effects (EPR), due to the high permeability of tumor-associated blood vessels [231]. Moreover, tumor-specific ligands can also be added to the surface of nanocarriers for active targeting of tumors [232]. Nano-sized delivery systems can penetrate physiological barriers, such as the blood–brain barrier, to improve drug delivery. In addition, the structure of nanocarriers allows the simultaneous encapsulation of multiple drugs, thus supporting combinatorial therapeutic strategies for overcoming ICI resistance.

In the era of personalized medicine, suitable biomarkers should be used to select the patient population who may benefit from the ICI-repurposed drug combination. It is noteworthy that the desired clinical efficacy of numerous repurposed drugs in cancer therapy was demonstrated only in a subset of patients [14]. A representative example is the remarkable enhanced patient survival benefit in colorectal cancer patients harboring tumors with low PD-L1 expression when they were treated with a combination of ICI and aspirin (Figure 6) [23]. A comprehensive analysis of molecular signatures, rather than a single gene mutation, is expected to be more useful in stratifying patient populations for specific ICI-repurposed drug combinations. Genome-wide association studies (GWAS) have been widely used to identify novel molecular targets for various diseases, but they may not be able to reveal the inherent complexity and heterogeneity of cancer. It works by identifying single nucleotide polymorphisms that are over-represented in a disease population versus a control population. In order to facilitate the personalized combination of ICI-repurposed drugs, advanced multifactorial data analysis methods are needed to unravel the convoluted association of molecular signatures driving the progression of specific cancer types. Recently, the ClinOmicsTrail^bc^ (a comprehensive visual analytics tool) has been used to analyze various genomics, epigenomics, and transcriptomics datasets to facilitate a comprehensive assessment of the use of repurposed drug candidates, immunotherapeutic agents, and targeted drugs for the treatment of breast cancer [22].

Furthermore, the emerging role of the gut microbiome in the clinical efficacy of ICI therapy has attracted a lot of attention [233]. The interaction between gut microbes and the repurposed drug can be complex and bidirectional. While the composition of the gut microbiome can be influenced by drugs, the gut microbiome can also affect an individual’s response to a drug by enzymatically changing the drug structure and altering its bioavailability [234]. This should also be taken into consideration when designing ICI-repurposed drug combinations for individual cancer patients.

## Figures and Tables

**Figure 1 pharmaceutics-15-02166-f001:**
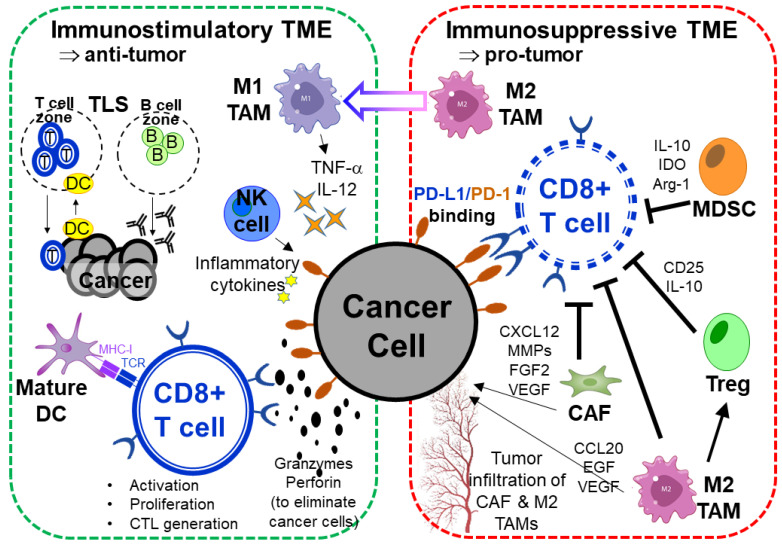
Regulation of the tumor microenvironment (TME) to control the efficacy of cancer immunotherapy. Immunostimulatory TME: Activated CD8^+^ cytotoxic T lymphocytes (CD8^+^ T cells), natural killer (NK) cells, mature dendritic cells (DC), and the anti-tumor M1 tumor-associated macrophages (M1 TAMs) elicit the anti-tumor immune response. Granzymes and perforin are secreted by cytotoxic T lymphocytes (CTL) and NK cells to directly eliminate cancer cells. M1 TAMs and NK cells could also secret tumor necrosis factor (TNF-α) and other proinflammatory cytokines to boost anti-tumor immunity. During persistent inflammation in tumors, immune cells migrated and clustered together to form secondary lymphoid organs known as “tertiary lymphoid structure (TLS)” [26]. TLS is a non-encapsulated lymph node-like structure that forms inside or adjacent to tumors. The microenvironment in TLS can allow better T-cell priming and prevent T-cell exhaustion. Circulating lymphocytes are recruited to tumor-associated TLS by lymphoid chemokines. Within the T-cell-rich areas of TLS, DCs take up tumor antigens and present processed antigens to specific T cells, thereby driving T-cell activation and differentiation. The effector T cells will migrate to the tumor sites for destruction of cancer cells. Within the B-cell follicle of TLS, tumor-infiltrating B cells are activated and mediate antibody production. Immunosuppressive TME: Cancer cells secrete chemokines and cytokines to recruit various immunosuppressive cells (including myeloid-derived suppressor cells (MDSC), regulatory T cells (Tregs), M2 tumor-associated macrophage (M2 TAM), and T helper 17 cells (Th17)) to maintain an immunosuppressive TME. These immunosuppressive cells suppress the cytotoxic functions of CD8^+^ T cells and NK cells via the expression and secretion of various factors (including CD25, indoleamine-2,3-dioxygenase (IDO), arginase 1 (Arg-1), etc.) to inhibit anti-tumor immune responses.

**Figure 2 pharmaceutics-15-02166-f002:**
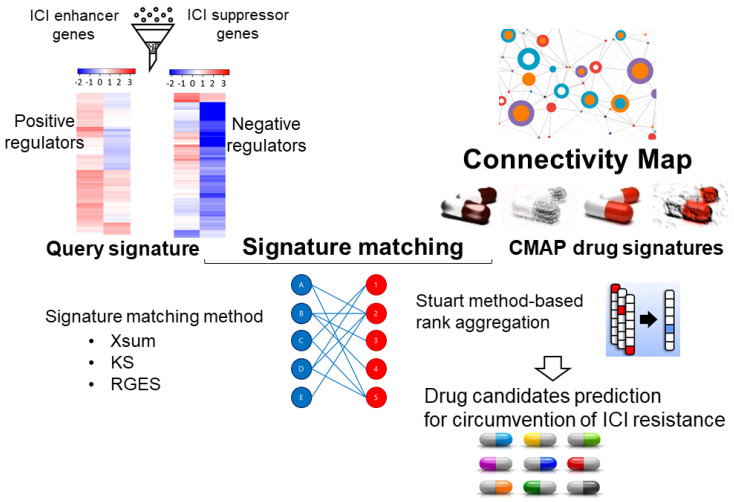
The computational workflow for the identification of repurposed drug candidates to overcome immune checkpoint inhibitor (ICI) resistance. A signature matching approach was adopted to identify potential combination partners that could produce a synergistic effect with ICIs. Using experimental data from ICI-treated cancer cells, potential regulators mediating resistance (ICI suppressor genes) or sensitivity (ICI enhancer genes) could be identified. Positive regulators are defined as the sensitizer genes, tumor immunity-related tumor suppressor genes, and ICI enhancer genes. Theoretically, upregulation of these positive regulators may give rise to enhanced anti-tumor immune response. A list of negative regulators was also similarly obtained. The positive and negative regulators collectively made up the query signature. On the other hand, the drug signatures (i.e., drug-induced profiles of gene expression changes) were downloaded from the Connectivity Map (CMap) datasets. Three signature matching methods, including eXtreme Sum (XSum), Kolmogorov–Smirnov (KS), and the Reverse Gene Expression Score (RGES), were used to match the query signature with the drug signatures. Finally, due to the different scales of the scores from the three methods, an order statistics-based method was used to integrate the results and generate a robust drug prediction result.

**Figure 3 pharmaceutics-15-02166-f003:**
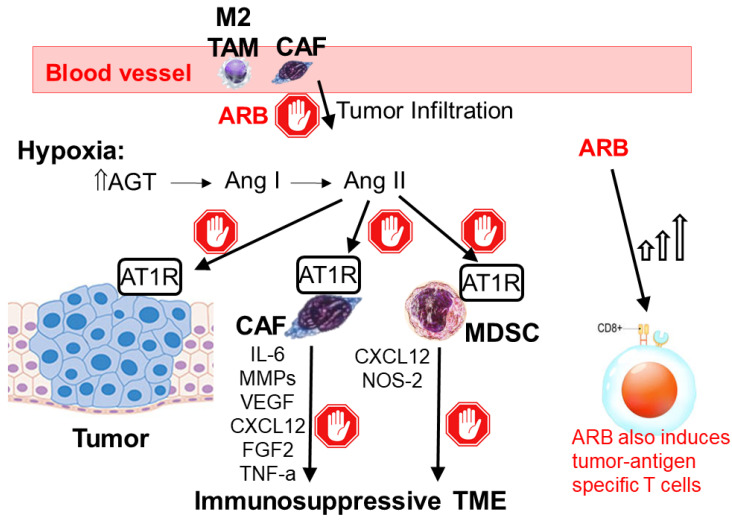
Mechanisms of angiotensin receptor blockers (ARBs) in overcoming ICI resistance. Angiotensinogen (AGT) is the precursor of all angiotensin peptides in the renin–angiotensin–aldosterone system, which plays a critical role in regulating blood pressure and fluid retention. The hypoxic tumor microenvironment is known to upregulate AGT. AGT is converted to angiotensin I (Ang I) and then to angiotensin II (Ang II) under physiological regulation. By inhibiting the binding of Ang II to the angiotensin type 1 receptor (AT1R) on tumor cells, ARBs suppress the tumor infiltration of cancer-associated fibroblast (CAF) and M2 tumor-associated macrophages (M2 TAMs). ARBs also retard the secretion of various immunosuppressive factors (IL-6, MMP2, and VEGF; and CXCL-12 and NOS-2) from CAF and myeloid-derived suppressor cells (MDSC), respectively, thus abolishing the immunosuppressive TME. Furthermore, ARBs also directly induce tumor antigen-specific T cells to stimulate an anti-tumor immune response.

**Figure 4 pharmaceutics-15-02166-f004:**
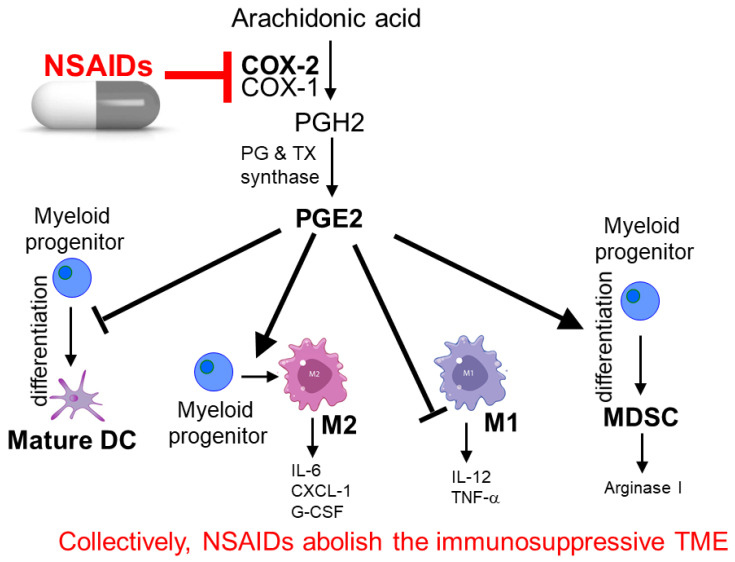
Mechanisms of NSAIDs to potentiate anti-cancer immunotherapy. NSAIDs inhibit cyclooxygenase (COX) enzymes (particularly COX-2) and synthesis of prostaglandin E2 (PGE2), subsequently retarding the release of various immunosuppressive factors, including IL-6, CXCL-1, and arginase I, from the pro-tumor M2 tumor-associated macrophages (TAMs) and myeloid-derived suppressor cells (MDSC) in the tumor microenvironment (TME). NSAIDs also activate tumoricidal M1 TAMs and give rise to type-I interferon (IFN-γ)-based anti-tumor immunity.

**Figure 5 pharmaceutics-15-02166-f005:**
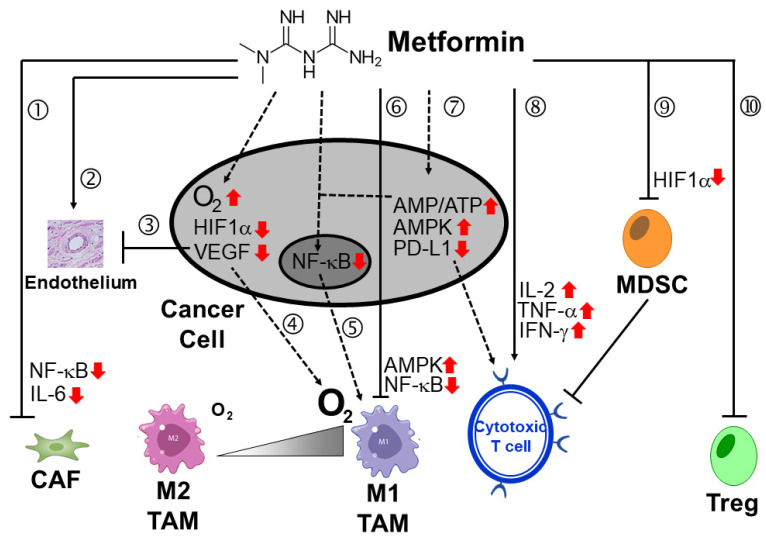
Schematic diagram showing the various mechanisms within the tumor microenvironment (TME) by which metformin potentiates the anti-tumor activity of cancer immunotherapy. ① Metformin suppresses tumor development by blocking cancer-associated fibroblast (CAF)-derived NF-κB pro-inflammatory signaling. ② Metformin could effectively reduce endothelial oxidative stress and improve endothelial function. ③ Metformin was reported to alleviate tumor hypoxia and reduce hypoxia-induced HIF-1α protein expression by promoting its degradation, which subsequently downregulated VEGF and suppressed angiogenesis. ④ The alleviation of tumor hypoxia by metformin could promote tumor-associated macrophage (TAM) repolarization towards the M1-tumoricidal phenotype. ⑤ Metformin may also activate AMPK and downregulate NF-κB to promote M1 TAM repolarization. ⑥ Metformin was reported to suppress macrophage inflammatory signals by activating AMPK but downregulating NF-κB. Metformin was shown to either decrease PD-L1 expression on cancer-cell surface ⑦, increase the number and promote the activity of cytotoxic T cell lymphocytes directly ⑧, or downregulate myeloid-derived suppressor cells (MDSC) to unlock cancer immunosuppression ⑨. ⑩ Metformin was also known to inhibit the pro-tumorigenic regulatory T cells (Treg). Direct and indirect effects of metformin are indicated by full and dashed lines, respectively. 
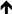
 = increased level; 
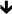
 = decreased level.

**Figure 6 pharmaceutics-15-02166-f006:**
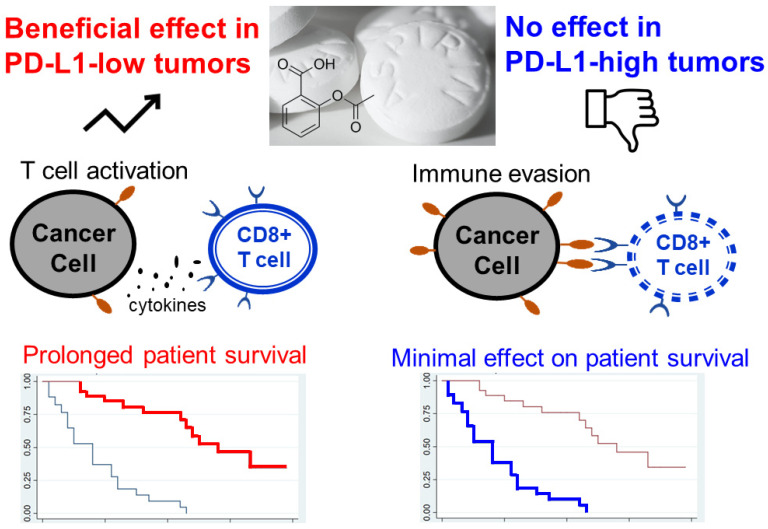
In clinical trials, aspirin was shown to produce patient survival benefit only in colorectal cancer patients bearing tumors with low PD-L1 expression. Left Kaplan-Meier survival curve: red line indicates prolonged survival in patients bearing PD-L1-low tumors. Right Kaplan-Meier survival curve: blue line indicates minimal effect (or inferior) survival in patients bearing PD-L1-high tumors.

**Table 1 pharmaceutics-15-02166-t001:** Representative clinical trials investigating the combination of PD-1/PD-L1 blockade therapy and various anti-cancer treatment modalities (ClinicalTrials.gov. Accessed 30 May 2023).

Drug Combination		Cancer Type	ClinicaTrials.gov Identifier (Phase)	Status
ICI	Other Treatment Modalities			
**Combination with chemotherapy**				
Alezolizumab (anti-PD-L1 mAb)	Carboplatin, Etoposide	Untreated extensive-stage SCLC	NCT04028050 (Phase 3)	Active; not recruiting
Atezolizumab (anti-PD-L1 mAb)	Pegylated liposomal doxorubicin, Cyclophosphamide	Metastatic triple-negative breast cancer	NCT03164993 (Phase 2)	Active; not recruiting
Nivolumab (anti-PD-1 mAb)	Lenalidomide (thalidomide analogue)	Relapsed or refractory non-Hodgkin or Hodgkin lymphoma	NCT03015896 (Phase 2)	Active; not recruiting
Anti-PD-1 mAb	Lenalidomide (thalidomide analogue) and azacytidine (epigenetic drug)	Relapsed/refractory peripheral T cell lymphoma	NCT05182957 (Phase 2)	Recruiting
Nivolumab (anti-PD-1 mAb), Ipilimumab (anti CTLA-4 mAb)	Trabectedin (marine derived and DNA-binding chemotherapeutic drug)	Advanced soft tissue sarcoma	NCT03138161 (Phase 2)	Recruiting
Pembrolizumab (anti-PD-1 mAb)	Metronomic cyclophosphamide	Metastatic breast cancer	NCT03139851 (Phase 2)	Completed; results not yet published
**Combination with targeted therapy**				
Atezolizumab (anti-PD-L1 mAb)	Cobimetinib (MEK inhibitor)	Previously treated unresectable locally advanced or metastatic CRC	NCT02788279 (Phase 3)	Completed; Median OS (8.87 months combination versus 7.10 months regorafenib); HR 1.00 (combination versus regorafenib) [69]
Atezolizumab (anti-PD-L1 mAb)	Cobimetinib (MEK inhibitor), Alectinib (ALK inhibitor), Entrectinib (ROS1 inhibitor), Vemurafenib (BRAF inhibitor), GDC-6036 (KRAS inhibitor)	Advanced or metastatic NSCLC (multiple trial arms including different combinations; estimated to recruit 1000 participants)	NCT03178552 (Phase 2/3)	Recruiting
Atezolizumab (anti-PD-L1 mAb)	Entinostat (Class I HDACI), Fulvestrant (Anti-estrogen), Ipatasertib (Akt inhibitor), Exemestane (steroidal aromatase inhibitor), Tamoxifen (SERM), Abemaciclib (CDK4/6 inhibitor)	HR-positive HER2-negative breast cancer	NCT03280563 (Phase 2)	Active; not recruiting
Avelumab (anti-PD-L1 mAb)	Axitinib (VEGFR, PDGFR, c-Kit inhibitor)	Advanced RCC	NCT02684006 (Phase 3)	Active; not recruiting
Carelizumab (anti-PD-1 mAb)	Apatinib (VEGFR, RET, c-Kit inhibitor)	Breast cancer	NCT04335006 (Phase 3)	Terminated (sponsor R&D strategy adjustment)
Ipilimumab (anti-CTLA-4 mAb); Nivolumab (anti-PD-1 mAb)	Cabozantinib (VEGFR2, Met inhibitor)	HCC	NCT01658878 (Phase 2)	Active; not recruiting
Nivolumab (anti-PD-1 mAb)	Ibrutinib (BTK inhibitor), Cetuximab (anti-EGFR mAb)	Metastatic HNSCC	NCT03646461 (Phase 2)	Active; not recruiting
Nivolumab (anti-PD-1 mAb)	Regorafenib (dual targeted VEGFR2-TIE2 TKI)	Gastro-oesophageal cancer	NCT04879368 (Phase 3)	Recruiting
Nivolumab (anti-PD-1 mAb)	Tivozanib (VEGFR, PDGFR, c-Kit inhibitor)	Renal cell carcinoma	NCT04987203 (Phase 3)	Recruiting
Pembrolizumab (anti-PD-L1 mAb)	Axitinib (VEGFR, c-Kit, PDGFR inhibitor)	Renal cell carcinoma	NCT02853331 (Phase 3)	Completed; median PFS (15.1 months combination versus 11.1 months sunitinib monotherapy) [70]
Pembrolizumab (anti-PD-L1 mAb)	Dasatinib (Abl, Src, c-Kit inhibitor), Imatinib mesylate (Abl, c-Kit, PDGFR inhibitor), Nilotinib (Bc-Abl inhibitor)	CML; patients with detectable minimal residual disease	NCT03516279 (Phase 2)	Recruiting
Pembrolizumab (anti-PD-L1 mAb)	Ibrutinib (BTK inhibitor)	Advanced colorectal cancer	NCT03332498 (Phase 1/2)	Completed; among 31 evaluable patients, 8 (26%) achieved stable disease; no objective response was observed [71]
Pembrolizumab (anti-PD-L1 mAb)	Letrozole (aromatase inhibitor), Palbociclib (CDK4/6 inhibitor)	Newly diagnosed metastatic stage IV ER-positive breast cancer	NCT02778685 (Phase 2)	Suspended (accrual on hold)—last update posted on 20 May 2023
Pembrolizumab (anti-PD-1 mAb)	Lenvatinib (VEGFR, FGFR, PDGFR, c-Kit, RET inhibitor)	Treatment naïve, metastatic NSCLC	NCT03829332 (Phase 3)	Active; not recruiting
Tislelizumab (anti-PD-1 mAb)	Sitravatinib (TAM family of receptors and VEGFR2 inhibitor)	Metastatic NSCLC	NCT04921358 (Phase 3)	Active; not recruiting
**Combination with epigeneticmodifying drugs**				
Pembrolizumab (anti-PD-1 mAb)	Azacitidine (DNA demethylating agent)	Pancreatic cancer	NCT03264404 (Phase 2)	Active; not recruiting
Pembrolizumab (anti-PD-L1 mAb)	Vorinostat (HDACI)	Stage IV NSCLC	NCT02638090 (Phase 2)	Active; not recruiting
Pembrolizumab (anti-PD-L1 mAb)	Vorinostat (HDACI), Tamoxifen (SERM)	Breast neoplasms	NCT02395627 (Phase 2)	Terminated; insufficient efficacy in an unselected patient population
Pembrolizumab (anti-PD-L1 mAb)	Decitabine (DNA demethylating agent), Radiation therapy	Pediatric and young adult cancer patients with solid tumor or lymphoma	NCT03445858 (Phase 2)	Active; not recruiting
**Combination with DNA damage response inhibitors**				
Atezolizumab (anti-PD-L1 mAb)	Niraparib (PARP inhibitor)	Recurrent ovarian cancer	NCT03598270 (Phase 3)	Active; not recruiting
Dostarlimab (anti-PD-1 mAb)	Niraparib (PARP inhibitor)	Metastatic endometrial or ovarian carcinoma	NCT03651206 (Phase 2/3)	Active; not recruiting
Pembrolizumab (anti-PD-1 mAb)	Olaparib (PARP inhibitor)	BRCA non-mutated advanced epithelial ovarian cancer	NCT03740165 (Phase 3)	Active; not recruiting
Pembrolizumab (anti-PD-1 mAb)	Olaparib (PARP inhibitor)	Unresectable, locally advanced NSCLC	NCT04380636 (Phase 3)	Recruiting
Pembrolizumab (anti-PD-1 mAb)	Olaparib (PARP inhibitor)	SCLC	NCT04624204 (Phase 3)	Recruiting
**Combination with Indoleamine 2,3-dioxygenase-1 (IDO1) inhibitors**				
Nivolumab (anti-PD-1 mAb)	Epacadostat (IDO1 inhibitor)	metastatic NSCLC	NCT03348904 (Phase 3)	Terminated (study halted prematurely and will not resume)
Pembrolizumab (anti-PD-1 mAb)	Epacadostat (IDO1 inhibitor)	urothelial cancer	NCT03361865 (Phase 3)	Completed; results not yet published
Pembrolizumab (anti-PD-1 mAb)	Epacadostat (IDO1 inhibitor)	metastatic RCC	NCT03260894 (Phase 3)	Active; not recruiting
Pembrolizumab (anit-PD-1 mAb)	Epacadostat (IDO1 inhibitor)	various solid cancers	NCT02178722 (Phase 1/2)	Completed; the combination was well tolerated and had encouraging anti-tumor activity in multiple advanced solid tumors. Objective responses in 12 (55%) of 22 patients with melanoma and other solid tumors [72]
**Combination with various other non-oncology drugs**				
Pembrolizumab (anti-PD-1 mAb)	COX inhibitor (aspirin or celecoxib)	MSI-H/dMMR or high TMB colorectal cancer	NCT03638297 (Phase 2)	Recruiting
Nivolumab (anti-PD-1 mAb)	COX-2 inhibitor (celecoxib)	Advanced “cold” solid cancers	NCT03864575 (Phase 2)	Not yet recruiting
Pembrolizumab or Nivolumab (anti-PD-1 mAb)	Antidiabetic drug (metformin or rosiglitazone)	Solid cancers	NCT04114136 (Phase 2)	Recruiting
Nivolumab (anti-PD-1 mAb)	Antidiabetic drug (metformin)	Stage III–IV NSCLC that cannot be removed by surgery	NCT03048500 (Phase 2)	Active; not recruiting
Nivolumab (anti-PD-1 mAb)	Antihypertensive drug—ARB (losartan)	Localized pancreatic cancer	NCT03563248 (Phase 2)	Active; not recruiting
Anti-PD-1 mAbs	Antidiabetic drug (metformin)	SCLC	NCT03994744 (Phase 2)	Recruiting

Abbreviations: ARB, angiotensin II receptor blocker; COX, cyclooxygenase; dMMR, mismatch repair deficient; ER, estrogen receptor; HDACI, histone deacetylase inhibitor; HER2, human epidermal growth factor receptor 2; HCC, hepatocellular carcinoma; HNSCC, head and neck squamous cell carcinoma; HR, hormone receptor; MSI-H, microsatellite instability-high; NSCLC, non-small cell lung cancer; OS, overall survival; PFS, progression-free survival; RCC, renal cell carcinoma; SCLC, small cell lung cancer; SERM, selective estrogen receptor modulator; TKI, tyrosine kinase inhibitor; TMB, tumor mutation burden.

## Data Availability

Not applicable.

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
