# Peer review of "Drug Repurposing to Circumvent Immune Checkpoint Inhibitor Resistance in Cancer Immunotherapy"

_pharmaceutics, 2023, doi:10.3390/pharmaceutics15082166_

Round 1

Reviewer 1 Report

Authors in this study they have given an overview of current research on drug repurposing to overcome ICI resistance. 

Although the work seems interesting, it is suggested that the manuscript should thoroughly checked for English language throughout.

Also, Fig. 1 or Fig. 2 etc should be removed from the respective images of the figures (it can be seen at the bottom of the each figure).

Other than this, this review article can be considered for publication in this journal.

Amending the above said comments are necessary. Upon which this manuscript be accepted for publication.

Author Response

Thank you so much for the positive comment and valuable suggestions from the reviewer.

The manuscript has been checked thoroughly and edited for English language.

The font “Fig #” have been removed from the figures.

Reviewer 2 Report

This review paper is well-organized and it is clear that the authors understand the field quite well. There are some points that need the attention of the authors.

Line 13: edit “immune”, should not be Bold.

Keywords: include “drug resistance: as well as other ICI.

Line 58-64: Please add Ref

Line 131: Please close parenthesis

-Regarding “Targeted drugs with anti-angiogenic activity” please be explicit on how this co-treatment could dominate resistance to ICI.

-Please refer to the role of phytochemicals in each section. There are various reports on the role of phytochemicals as immune-potentiators which has been used in combination with ICI. Related references include DOI: 10.3390/cancers15030653, 10.1016/j.biopha.2022.113618

-I suggest discussing the role of microbiota in regulating cancer immunometabolism.

-I also suggest addressing the combination of ICI with nano-particluate delivery vehicles encapsulating different agents.

-Please omit p value from the text, usually we do not report them in the review paper.

-Line 407 and 409: Please omit Ref 108, also omit Ref 109 in line 415, and Ref 111 in line 424. Also Ref 120 in lines 470 and 475. Line 564, Ref 144. And Ref 145 in Line 569. Please carefully check the text to omit repeated references. There are many repeated references through the text.

-I suggest revising the font size in the figures. We usually use different font size to make it easier to understand the figure. When looking at the figure, all fonts seems the same, so it’s difficult to recognize which text is prior to others.

-Figure 4. Please reorder numbering in this figure and the caption, respectively, from left to right, 1, then 2 and 3, first 7, then 8.

-Regarding “Targeting amino acid catabolism to potentiate cancer immunotherapy” please also address the glutamine addiction of cancer cells and the role of its antagonist in potentiating tumor immunity and relevant studies, or its potential co-combination therapy with ICI. I refer you to DOI: 10.1126/science.aav2588 and similar studies.

-I suggest locating the section 4 before section 3, means first introduce the methods to choose the candidate drugs and then introduce the drugs used for combination therapy.

-Figure 6. Please add the permission code if this figure is not original.

-Challenges and Perspectives: Please also highlight the role of nanotechnology in ameliorating the combination therapy with ICI.

Needs minor editing

Author Response

Reviewer 2 Comments:

This review paper is well-organized and it is clear that the authors understand the field quite well. There are some points that need the attention of the authors.

Line 13: edit “immune”, should not be Bold.

Keywords: include “drug resistance: as well as other ICI.

Response:

Thank you so much for the positive comment and valuable suggestions from the reviewer.

The font has been corrected to regular font (line 18).

“Drug resistance” and “ICI” have been added as keywords.

Line 58-64: Please add Ref

Response:

Relevant references have been added to the paragraph (reference #5-10)

Line 131: Please close parenthesis

Response:

Correction has been made accordingly.

-Regarding “Targeted drugs with anti-angiogenic activity” please be explicit on how this co-treatment could dominate resistance to ICI.

Response:

A brief section is added to explain the relevance of angiogenesis to ICI resistance (line 330-336).

“Cancer angiogenesis is a critical process that facilitates the formation of new and abnormal blood vessels to support tumor growth and metastasis. Effective antitumor immune response requires a series of events including activation of T cells, recruitment of immune cells, and recognition and subsequent killing of cancer. To this end, inducers of angiogenesis are known to interfere with activation, infiltration and function of T cells. Moreover, the tumor vasculature is known to promote an immunosuppressive TME, which can be reversed by anti-angiogenic therapies”   

-Please refer to the role of phytochemicals in each section. There are various reports on the role of phytochemicals as immune-potentiators which has been used in combination with ICI. Related references include DOI: 10.3390/cancers15030653, 10.1016/j.biopha.2022.113618

Response:

The specific role of each phytochemicals in overcoming ICI resistance is summarized for each example (line 928-929, 933, 948-949, 952, 958-959, 960-961, 965). The review article about new advances of phytochemicals in tumor immunotherapy (Liu et al, 2022; 10.1016/j.biopha.2022.113618) is cited in the manuscript.

-I suggest discussing the role of microbiota in regulating cancer immunometabolism.

Response:

A brief section is added to discuss the role of microbiota in regulating cancer immunometabolism (line 968-981).

“The gut microbiota refers to the collection of numerous microbes inhabiting the human intestinal epithelium in the gastrointestinal tract. The microbiota maintains complex interactions with the intestinal cells and resident immune cells through their metabolites. Short-chain fatty acids, purine metabolites, and tryptophan derivatives are well-known gut microbiota-derived metabolites that regulate immune responses and modulate antitumor efficacy of ICI (219,220).”

-I also suggest addressing the combination of ICI with nano-particluate delivery vehicles encapsulating different agents.

Response:

A brief section is added to discuss the use of nano-particle formulation to deliver different ICI agents and their combination (line 1121-1131).

“Nanoparticle (NP)-based formulations have been used to deliver ICIs with an aim to increase specificity, decrease toxicity, and potentiate the immunostimulatory effect [226]. NPs could be designed, with different size, geometry and composition, to facilitate the localized and controlled ICI release, and to protect ICI stability after administration into the body. Nanocarriers are preferentially delivered into tumors by passive targeting via enhanced permeation and retention effects (EPR), due to the high permeability of tumor-associated blood vessels [227]. Moreover, tumor-specific ligands can also be added to the surface of nanocarriers for active targeting of tumors [228]. Nano-sized delivery systems can penetrate physiological barriers, such as the blood brain barrier, to improve drug delivery. In addition, the structure of nano-carriers allows simultaneous encapsulation of multiple drugs, thus supporting combinatorial therapeutic strategies for overcoming ICI resistance.”  

-Please omit p value from the text, usually we do not report them in the review paper.

Response:

P value is removed from the text.

-Line 407 and 409: Please omit Ref 108, also omit Ref 109 in line 415, and Ref 111 in line 424. Also Ref 120 in lines 470 and 475. Line 564, Ref 144. And Ref 145 in Line 569. Please carefully check the text to omit repeated references. There are many repeated references through the text.

Response:

Repeated references have been removed throughout the manuscript.

-I suggest revising the font size in the figures. We usually use different font size to make it easier to understand the figure. When looking at the figure, all fonts seems the same, so it’s difficult to recognize which text is prior to others.

Response:

Font size has been revised accordingly in the figures.

-Figure 4. Please reorder numbering in this figure and the caption, respectively, from left to right, 1, then 2 and 3, first 7, then 8.

Response:

The numbering in the figure (Figure 5 in the revised manuscript) has been revised accordingly.

-Regarding “Targeting amino acid catabolism to potentiate cancer immunotherapy” please also address the glutamine addiction of cancer cells and the role of its antagonist in potentiating tumor immunity and relevant studies, or its potential co-combination therapy with ICI. I refer you to DOI: 10.1126/science.aav2588 and similar studies.

Response:

A brief section about glutamine addiction of cancer cells and the use of glutamine antagonist to boost ICI has been added (line 883-912).

“Glutamine is an important source of carbon and nitrogen necessary for anabolic metabolism of cancer cells [199]. As glucose is metabolized to lactic acid by glycolysis in cancer cells, glutamine is used to generate metabolic intermediates via the tricarboxylic acid (TCA) cycle to support cancer metabolism and proliferation. This specialized metabolism of cancer cells creates acidic, hypoxic, nutrient-depleted, and immunosuppressive TME that is opposing antitumor immune responses. Therefore, the inhibition of glutamine metabolism is expected to inhibit cancer growth and also restore antitumor immunity. In fact, accumulating evidence suggests that inhibitors of glutamine metabolism, such as V-9302 and 6-diazo-5-oxo-L-norleucine (DON) derivatives, give rise to pronounced antitumor effect when used in combination with ICIs by remodeling the immunosuppressive TME [200,201]. In murine models, V-9302 was shown to induce tumoral PD-L1 expression and augment immune evasion [202]. The blockade of glutamine metabolism reduced cellular GSH level and endoplasmic reticulum Ca2+-ATPase (SERCA) glutathionylation, subsequently leading to a reduced SERCA activity. The elevation of cytosolic Ca2+ level activated calcium/calmodulin-dependent protein kinase II, thereby leading to aberrant NF-kB signaling and higher PD-L1 expression. On the other hand, DON derivatives inhibit a broad range of glutamine-requiring enzymes including glutaminase [201]. Effector T cells responded to DON derivatives by upregulating oxidative metabolism and adopting a highly activated phenotype, thereby giving rise to higher antitumor immunity.”

-I suggest locating the section 4 before section 3, means first introduce the methods to choose the candidate drugs and then introduce the drugs used for combination therapy.

Response:

We have reordered section 3 and 4 accordingly.

-Figure 6. Please add the permission code if this figure is not original.

Response:

Figure 6 is prepared by the authors for this manuscript. There is no copyright issue.

-Challenges and Perspectives: Please also highlight the role of nanotechnology in ameliorating the combination therapy with ICI.

Response:

A brief section is added to discuss the use of nano-particle formulation to deliver different ICI agents and their combination (line 1121-1131).

“Nanoparticle (NP)-based formulations have been used to deliver ICIs with an aim to increase specificity, decrease toxicity, and potentiate the immunostimulatory effect [226]. NPs could be designed, with different size, geometry and composition, to facilitate the localized and controlled ICI release, and to protect ICI stability after administration into the body. Nanocarriers are preferentially delivered into tumors by passive targeting via enhanced permeation and retention effects (EPR), due to the high permeability of tumor-associated blood vessels [227]. Moreover, tumor-specific ligands can also be added to the surface of nanocarriers for active targeting of tumors [228]. Nano-sized delivery systems can penetrate physiological barriers, such as the blood brain barrier, to improve drug delivery. In addition, the structure of nano-carriers allows simultaneous encapsulation of multiple drugs, thus supporting combinatorial therapeutic strategies for overcoming ICI resistance.”  

Reviewer 3 Report

To and Cho provide an informative review on the repurposing of in clinic drugs to improve the anti-tumor efficacy of immune checkpoint blockade. Agents for inclusion in combination ICI-based protocols include chemotherapeutic agents, anti-angiogenic drugs (TKIs, anti-VEGFRs), CDKis, DDRis, DNMTis/HDACis, and “repurposed” drugs such as ARBs, NSAIDs, modulators of cellular metabolism, inhibitors of acidosis/hypoxia, IDOi and modulators of the gut microbiome (including TCM), with appropriate discussion of anticipated immunologic mechanisms of action. High-throughput screening methods (both laboratory and virtual) are described for empirical identification of novel agents. Issues related to drug efficacy differences based on cancer type as well as individual patient characteristics are discussed. The Figures and Table supplied are both informative and well-described. The cited literature is comprehensive and up-to-date. Overall, this is a useful primer/update to those interested in an expanded repertoire of cancer immunotherapeutic regimens that may improve patient response to ICI.       

My only minor concern reflects recent clinical interest in tertiary lymphoid structures (non-encapsulated lymph node-like structures that form in tumor sites) and their association with improved response to ICI. In this light, Fig. 1 and the general discussion of the tumor immune microenvironment should be modified to include (mature) B cells/B cell immunobiology that are strongly correlated with improved patient outcomes on-treatment with ICI.

Minor grammatical errors requiring inspection/correction.

Author Response

Response:

Thank you so much for the positive comment and valuable suggestion from the reviewer.

Figure 1 has been revised accordingly to describe the involvement of tertiary lymphoid structure (TLS) in antitumor response to ICI.  

Round 2

Reviewer 2 Report

No comments. well done

Moderate